# The Trace Gas Monitoring Method Based on Diode Laser Wavelength-Modulation Spectroscopy Technology for the Detection of Clinical Blood Infection

Jing Sun [1,2], Yuxiao Song [2], Dongxin Shi [2,3], Feifei Wang [2], Yong Yang [2], Pengyu Yao [2], Binghong Song [2], Yang Yu [4], Chenyu Jiang [1,2,3,*] and Bingqiang Cao [1,*]

1. School of Materials Science and Engineering, University of Jinan, Jinan 250022, China; 18253121729@163.com
2. Jinan Guoke Medical Technology Development Co., Ltd., Jinan 250001, China; songyuxiao@sibet.ac.cn (Y.S.); shidx@sibet.ac.cn (D.S.); wangffqd@163.com (F.W.); yangyong@sibet.ac.cn (Y.Y.); yaopy@sibet.ac.cn (P.Y.); songbh@sibet.ac.cn (B.S.)
3. Suzhou Institute of Biomedical Engineering and Technology, Chinese Academy of Sciences, Suzhou 215163, China
4. Shandong Engineering Consulting Institute (Shandong Provincial Government Investment Project Evaluation Center), Jinan 250014, China; 18653184597@163.com
* Correspondence: jiangcy@sibet.ac.cn (C.J.); mse_caobq@ujn.edu.cn (B.C.)

**Abstract:** It is important to monitor and evaluate the growth of microorganisms in order to accurately judge the situation of blood microbial infection. In this paper, diode laser wavelength modulation spectroscopy (DLWMS) technology is used to design a set of low-cost, high sensitivity, fast dynamic responses and a non-invasive trace gas measurement system, which can quickly and accurately assess the concentration of carbon dioxide ($CO_2$) produced by blood microbial reproduction. The measurement principle and spectral processing algorithm of DLWMS are introduced first. The automatic and rapid detection of $CO_2$ is realized through a self-designed optical system. By using the system to detect blood infection, the accuracy of the technology was verified. Therefore, it also indicates that DLWMS $CO_2$ monitoring is a highly sensitive, fast-response and non-invasive technology, which can accurately and quickly determine blood infection and meet the clinical application requirements of human septicemia, bacteremia and other diseases.

**Keywords:** diode laser wavelength modulation spectroscopy (DLWMS); $CO_2$ monitoring; blood infection





## 1. Introduction

A blood culture test is a microbiological test to check for the presence of bacteria in blood samples. When a patient shows symptoms of an associated infection, blood culture results can identify the type of microbe causing the infection. Normal blood is sterile; bacterial invasion can lead to bacteremia, sepsis, sepsis, etc. so timely and accurate etiological diagnosis is important [1–4]. According to reports, the mortality rate of critically ill patients after infection complications is as high as 25%. Hence, rapid and accurate detection of bacterial infection in the blood is very important for rational drug use, effective treatment and reduction of patient mortality, and is the key to effective clinical treatment.

At present, the traditional methods of microbial growth detection mainly include dry weight measurement [5], volume measurement [6], the turbidimetric method [7] and the physiological index method [8]. Due to their versatility and simple operation, they have been widely used in experiments. However, the traditional methods take a long time to operate and require a lot of manual operation, which makes it impossible to use this technology for automated measurement. Invasive operations can cause bacterial interference, resulting in a loss of accuracy. Therefore, the traditional method has a low positive rate, long culture cycle, is easily polluted, and has difficulty meeting clinical needs. Therefore, there is a desperate need for a detection technique to quickly identify microbial growth, which is of great significance for early antibiotic treatment in patients.

In recent years, traditional detection methods have been unable to meet the requirements of the rapid detection and analysis of microorganisms due to their own shortcomings. Some new methods, such as the physiological index, have been widely applied to detect microbial growth [9,10]. Among them, the amount of $CO_2$ produced in the process of bacterial reproduction and metabolism changes rapidly with time, thus reflecting the state of bacterial growth. In the study of microbial growth, the technology of $CO_2$ detection based on colorimetric analysis is quite mature [11–13]. For instance, commercial BacT/ALERT 3D automated blood culture systems are used to monitor blood culture status in the clinic. However, this method takes a long time to obtain a data point and, in the detection process, the $CO_2$ sensing layer in the culture bottle is easily contaminated by bacteria. In addition, the response time of chromatography is long, which can reduce accuracy in long-term testing.

Compared with colorimetry and chromatography, diode laser wavelength modulation spectroscopy (DLWMS) technology uses the characteristic absorption line of the substance to be measured to measure the concentration. Different spectral frequencies are selected, the absorption of gas will have a great change, and the spectral frequency has a significant impact on the linear function of the absorption spectrum. Therefore, as long as the appropriate absorption line is selected, this technology will have the advantages of high sensitivity and fast dynamic response, and has been widely used in many fields [13–21]. The detection method based on spectral technology was first successfully applied to the determination of microbial growth curves by Shao et al. in 2016 [22]. Based on this, we developed and designed a spectral high-throughput trace gas measurement system based on DLWMS technology for the detection of blood microorganisms, which can directly, quickly and accurately determine the $CO_2$ concentration level in the metabolic process of blood microorganisms. It is a non-invasive, highly sensitive and real-time monitoring technology.

## 2. Materials and Methods

### 2.1. Principle and Method of Measurement

Absorbance $\alpha(v)$ is defined as the ratio of the initial and transmitted light intensity of a laser. It is defined as follows (Equation (1)):

$$a(v) = \ln\left(I_{0(v)}/I_{(v)}\right) \tag{1}$$

Equation (1) is called Lambert–Beer's law. The fundamentals of WMS can be found in References [16,22]. In this paper, the absorption spectrum signals obtained in the experiment are processed in three steps. Firstly, the waveforms of $1f$ and $2f$ are calculated, and the original signals of $1f$ and $2f$ are fitted. When the absorbance is low, we can write it as Equation (2).

$$H_k(V,\,V_a) = -2PXSLf \int_{-1/2f}^{1/2f} \varnothing[V + V_a\cos(2\pi ft)]\cos(2\pi kft)dt \tag{2}$$

where $V$ is the center frequency of the laser, $V_a$ is the amplitude of modulation, $f$ is the modulation frequency, $\varnothing(v)$ is the function of area normalized, $P$ is the total pressure, $X$ is the mole fraction of the absorbed substance, and $S$ and $L$ are the line strength of the target line and the path length, respectively. $H_k(V,\,V_a)$ is the coefficient of the $k$th-order Fourier. When the value of $k$ corresponds to 1 or 2, $H_k(V,\,V_a)$ equals the $1f$ or $2f$ signals, respectively [23–26].

Since the largest harmonic whose resonance center is not zero is the second harmonic component, WMS is usually implemented by detecting the second harmonic component. However, in order to eliminate the influence of common factors such as laser output intensity, photoelectric gain and laser transmission changes, the amplitude of $2f$ signal and

$1f$ signal can be normalized [27–29]. This means that the growth of microbes in the blood is directly proportional to the amount of $CO_2$ produced, and $X$ can be written as:

$$X \propto \frac{H_2(v_0,\ v_a)}{H_1(v_0,\ v_a)} \tag{3}$$

### 2.2. Selection of $CO_2$ Absorption Spectra

According to the formation mechanism of the gas absorption spectrum, gas molecules absorb light waves of different frequencies differently. Therefore, continuous absorption bands will be formed when continuous light waves are used to pass through the gas to be measured, as shown in Table 1.

**Table 1.** Absorption bands of major atmospheric components.

|  | Strong Absorption | | Weak Absorption | |
|---|---|---|---|---|
|  | Wavelength μ/m | Wave number/cm$^{-1}$ | Wavelength μ/m | Wave number/cm$^{-1}$ |
| $H_2O$ | 1.4<br>1.9<br>2.7<br>6.3<br>13.0–100 | 7142<br>5263<br>3704<br>1595 | 0.9<br>1.1 | 11,111<br>9091 |
| $CO_2$ | 2.7<br>4.3<br>14.7 | 3704<br>2320<br>680 | 1.4<br>1.6<br>2.0<br>5.0<br>9.4<br>10.4 | 7142<br>6250<br>5000<br>2000<br>1064<br>962 |
| $O_3$ | 4.7<br>9.6<br>14.1 | 2128<br>1042<br>709 | 3.3<br>3.6<br>5.7 | 3030<br>2778<br>1754 |
| $CH_4$ | 3.3<br>3.8<br>7.7 | 3030<br>2632<br>1299 |  |  |

On-line detection of $CO_2$ based on tunable diode laser absorption spectroscopy (TD-LAS) and wavelength modulation spectroscopy (WMS) is the most common non-extraction technology. According to the absorption band of $CO_2$ summarized in Table 1, considering the detection sensitivity of the selected optical path, the interference of water and gas, as well as the technology and cost of a semiconductor laser corresponding to the band, it is meaningful to select the absorption line of a 2.0 μm attachment to detect $CO_2$ concentration level. This is the best way to greatly reduce the cost on the basis of ensuring the sensitivity of detection.

In the clinical detection of blood infection, the concentration of $CO_2$ in the blood culture bottle increases gradually due to the reproduction and metabolism of bacteria, resulting in the total pressure in the bottle rising to between 1 and 1.5 atmospheres. At the operating temperature of 37 °C, the moisture content is almost saturated, which is the main interference factor of the $CO_2$ spectrum. According to relevant data from the HITRAN database [14], after comparing the absorption lines of common gas molecules such as $CO_2$, water vapor and CO near 2.0 μm, the characteristic absorption line with a wavelength of 2004 nm was finally selected as the absorption wavelength for measuring $CO_2$ concentration, which can effectively improve detection sensitivity and selectivity.

### 2.3. Design of Optical Detection System Based on DLWMS Technology

The device of a trace gas optical detection system based on DLWMS technology is shown in Figure 1. In this system, a semiconductor laser is used as the laser source, and the

sawtooth wave and sine wave signals generated by the signal generator are superimposed on the controller of the laser to realize the tunable wavelength of the laser. A 2004 nm tunable distributed feedback (DFB) diode laser with an output power of 10 mW is applied to the light source. The central wavelength of the semiconductor laser is set as 4991.26 $cm^{-1}$ through the laser controller. The optical path is adjusted and calibrated to pass through the culture flask, and the length of the optical path is 3 cm. Finally, the optical path is focused on the photodiode detector, and the signals collected are processed by an efficient data analysis and processing system to achieve fully automated detection of blood infection microorganisms.

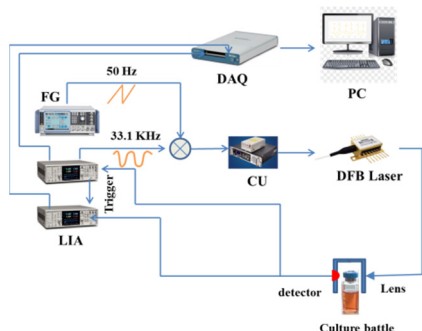

**Figure 1.** Schematic diagram of optical detection system based on DLWMS technology.

## 2.4. Design of Spectroscopic High Throughput Blood Culture Apparatus

The spectroscopic high-throughput blood culture instrument based on DLWMS trace gas optical detection technology is mainly composed of four parts: detection module, storage module, control module and information acquisition and processing module. The 3D model of its mechanical structure is shown in Figure 2.

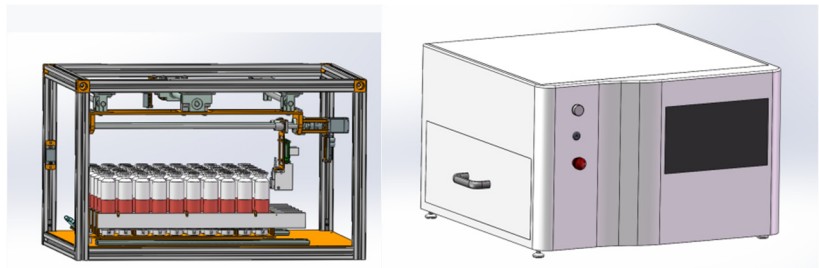

**Figure 2.** Three dimensional (3D) model of the mechanical structure of spectroscopic high throughput blood culture apparatus.

As can be seen from Figure 1, all culture bottles are placed in a 10 × 6 square storage module, and sensors are installed at each bottle position. When the culture bottles are placed in the bottle position, their position and time information will be transmitted to the upper computer. The module is installed in the drawer, which is fixed by the slide rail and can be taken out automatically or manually in general. In order to ensure the stability of the optical path, in the detection module the positions of the laser and the detector are consistent with each other by mechanical structure. In the detection process, the module will complete the detection of all culture bottles as a whole with the moving trajectory of the stepper motor. The control module mainly realizes position control, culture temperature control and mixing control. The position control module determines the moving trajectory of the detection module. Temperature control is mainly achieved by sticking a heating plate at each row of bottle positions. On the basis of uniform heating, the temperature of each culture bottle is guaranteed to be constant. In the temperature control module, the temperature of each culture flask is detected and alerted by a hyperthermocouple sensor, and the bottle temperature can be controlled within 0.5 °C. In the mixing part,

the self-made culture bottle with magnetons and the magnetic stirrer at the bottom of the bottle are used to realize the mixing of culture medium and samples and promote the rapid growth of bacteria. When a particular culture bottle is tested, the information collection and processing module will identify the location of the bottle and display its position through the LCD and alarm.

## 3. Results and Discussion

### 3.1. Debugging and Performance Analysis of Optical Inspection System Based on DLWMS Technology

According to the basic principle of DLWMS and related spectral processing methods, the second harmonic debugging of $CO_2$ gas with 10% concentration was carried out based on the ambient air, and the results are shown in Figure 3. The results show that, under the optical path conditions of the equipment, 10% of $CO_2$ gas shows a good second harmonic signal, which can provide a basis for a follow-up highly sensitive detection of blood infection.

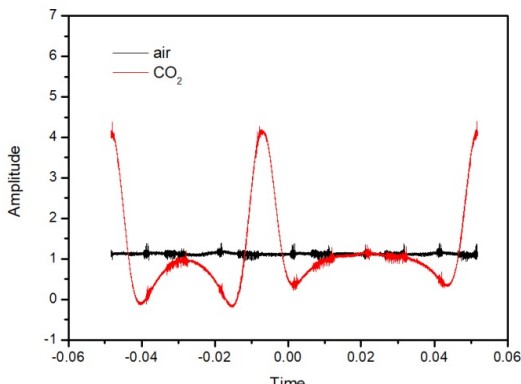

**Figure 3.** Second harmonic signal of spectroscopic high throughput blood culture apparatus.

The stability of the instrument was tested and verified after the optical path calibration and debugging was completed and the second harmonic signal was obtained. In order to ensure the accuracy of detection and promote the growth and reproduction of bacteria in the culture bottle, it was necessary to maintain a constant growth temperature of 37 °C and constant mixing during the culture process, so that bacteria could fully contact the culture medium. Therefore, the stability of the temperature control (Figure 4) and agitation mixing (Figure 5) of the equipment was tested. The test results of temperature control stability are shown in Figure 4. As we can see, the temperature control unit of the equipment can keep the temperature of each culture bottle constant at about 37 °C on the basis of uniform heating, and the fluctuation range of the temperature is less than 0.5 °C, which has good temperature control stability. Figure 5 shows the signal stability test results of the equipment under the conditions of stirring and mixing. The amplitude of the second harmonic wave floats at 2.3 mV, within the allowable range of error. This indicates that the mixing of bacteria and culture medium in the process of culture by magnetic stirring not only has a good mixing effect, but also does not cause the fluctuation of the test signal.

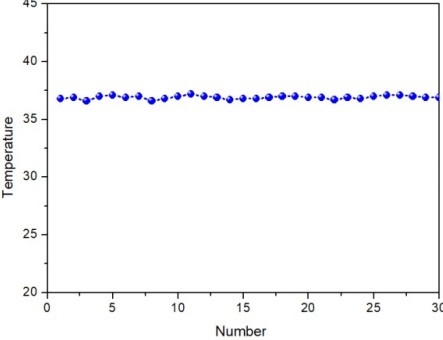

**Figure 4.** Temperature distribution curve of partial culture flask.

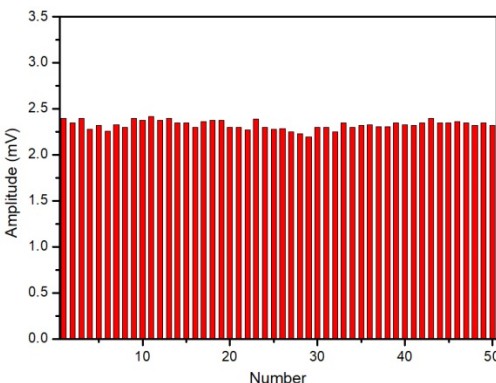

**Figure 5.** Signal stability curve in mixing process.

On the basis of good stability test of the instrument, standard $CO_2$ gas with concentrations of 5%, 10%, 15% and 20% was used to calibrate the instrument (Figure 6). It is shown that $CO_2$ gas with a concentration gradient of 5% can be clearly distinguished according to the second harmonic amplitude, and the fitting results also show that there is a good linear relationship between the second harmonic amplitude and $CO_2$ gas concentration (average value of 5000 tests).

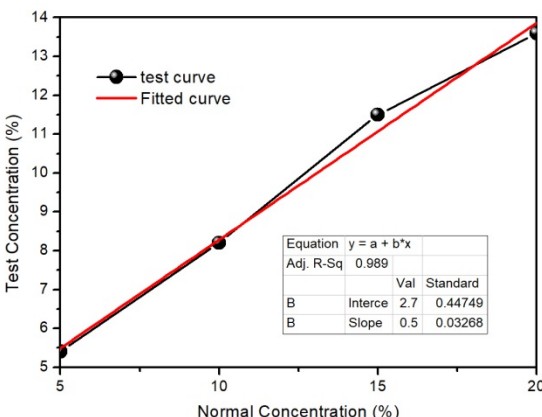

**Figure 6.** Linearity curve of spectroscopic high throughput blood culture apparatus.

### 3.2. Preliminary Study of the Clinical Application

In order to test the accuracy of the spectrometric high-throughput blood culture instrument based on DLWMS trace gas detection technology, the optical detection system based on DLWMS and the commercial system were used to test each blood sample separately. It is well known that the metabolism of blood-infected bacteria causes changes in the level of $CO_2$ concentration in culture vials. Some of this $CO_2$ comes from the growth of normal red blood cells in the body. Therefore, it is crucial to set a threshold to determine $CO_2$ concentrations and thus the presence of bacteria [22]. According to previous studies [22], the threshold value of aerobic culture bottle is set at 0.15 arbitrary unit, and that of the anaerobic culture bottle is set at 0.055 arbitrary unit. Thus, when the resulting voltage signal is greater than the threshold, the bottle is recognized as a positive result, and vice versa. Under the condition of strict control of variables, we preliminarily tested 50 samples of aerobic blood culture for a total of 7 days. In order to verify the accuracy of the results of the optical detection system based on DLWMS, culture vials were not removed from the DLWMs-based optical detection system until they were identified as having positive results in the commercial system or arrived 7 days later, and the comparison results are shown in Figure 6.

Figure 7 shows the culture results of samples from suspected infected patients in aerobic culture bottles. The results are marked with different colors. The height of the line of the positive result represents the signal amplitude of the DLWMS optical detection

system when the commercial equipment recognizes the positive result or negative result. The results show that the negative results measured by the new optical detection system based on DLWMS are all below the threshold value, and the positive results are all above the threshold value, without false negatives or false positives, which is highly consistent with the results measured by the commercial equipment.

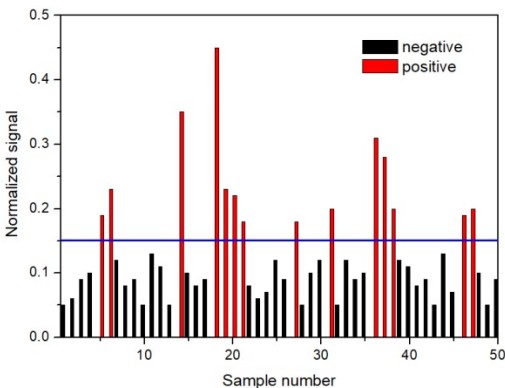

**Figure 7.** Results of aerobic culture bottles.

## 4. Conclusions

The trace gas detection system based on DLWMS technology independently designed and developed in this paper has the advantages of being low-cost, high sensitivity, non-invasive and with a fast dynamic response time, and is used to detect $CO_2$ concentration level during the multiplication of bacteria in the blood, so as to monitor microbial growth. The experimental results show that the results of the optical detection system based on DLWMS are consistent with those of the commercial system, but greatly shorten the positive sample detection time to some extent. Therefore, the new blood culture optical detection system based on DLWMS can judge the infection situation of patients earlier and improve the rescue time of patients. The self-developed trace gas optical detection system based on DLWMS can be used for rapid and accurate microbial growth analysis in hospitals. For future development, it has a broad application prospect.

**Author Contributions:** Conceptualization, J.S.; methodology, F.W.; software, Y.S.; validation and formal analysis, D.S.; investigation, P.Y.; resources, C.J. and B.C.; data curation, Y.Y. (Yong Yang) and Y.Y. (Yang Yu); writing—original draft preparation, J.S.; writing—review and editing, B.S. All authors have read and agreed to the published version of the manuscript.

**Funding:** This work is supported by the National Natural Science Foundation of China (61805273), the key project of "Twenty Universities" in Jinan which is "Development and Application of Cavity Ring-down Spectroscopy System for Efficient Detection of Breath Markers" (2019GXRC043), and the Innovation talents project of Quancheng "5150" Talent multiplication Plan "Research and development and industry of breathing gas online analyzer based on highly sensitive laser spectrum technology".

**Institutional Review Board Statement:** Not applicable.

**Informed Consent Statement:** Not applicable.

**Data Availability Statement:** The data that support the findings of this study are available from the corresponding author upon reasonable request.

**Conflicts of Interest:** The authors declare no conflict of interest.

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
