# Peer review of "The Trace Gas Monitoring Method Based on Diode Laser Wavelength-Modulation Spectroscopy Technology for the Detection of Clinical Blood Infection"

_processes, doi:10.3390/pr10081450_

Round 1
Reviewer 1 Report
The paper is a quite interesting application of a common inspection technique (e. g. for oxygen in pharmaceutical products, vials) to the analysis of carbon dioxide in blood samples. The method itself is not new but the novelty is in the application.
A few comments on the text.
- The english could be improved. Consider a global review.
- 112. Can you include some more details, for example just the simulated spectra, to show why the 2004nm region has been chosen to improve detection sensitivity and selectivity?
- 130. Fig. 1. "Culture bottle" (or maybe vial). Can you improve the drawing to show the optical layout in a more detailed view? Also, what are the two LIAs? Usually in the kHz range the signal is sampled and digital lock-in detection is performed. It is not clear why the 33.1kHz modulation frequency appears to be generated by the LIA.
- 137 and also in Fig. 3. What does mean "high flux"? Maybe "high throughput". If so, how fast the single CO2 measurements are performed?
- 139. Which kind of sensor?
- 159 and later in the text. I would use the terms "calibration" or "validation" rather than "debugging", which carries the idea of something wrong with the instrument.
- 168. The "relative wavenumber" scale is wrong, as the spectral features should then be symmetrical to the center point. I know there are several reasons (tuning delays) to explain this phenomenon, but they should either explained in the text or a different wording for the x-axis should be used.
- 187. Can you put english wording in the graph?
- 195. Also here, check the language of the graphs. Can you add an error bar on the right-side graph?
- 200. Please define the "commercial system". Which are the reference methods to tell if a sample is positive? Which method did you use in the study?
- 205. Please explain the definition and meaning of the arbitrary units.
- 215. Do you have also time-resolved data on the CO2 level evolution during the 7-days incubation time?
- 226. "trace gas" maybe is not appropriate. Can you convert the threshold value you used in actual CO2 vol. % units?
- 231. Time-resolved measurement should be reported if you want to adequately support the statement about the reduction in positive detection time.
Author Response
We appreciate the reviewer’s favorable reviewing and recognition of this manuscript. And we thank the reviewer for the constructive comments.Please refer to the attached document for details. Thank you very much!

Reviewer 2 Report
This article presents an instrument for automatic detection of clinical blood infections. I believe that the instrument is a quite useful tool and might be even utilize in practice in medical diagnostics. The results prove that it really works, not worse than commercially available counterparts. From the scientific point of view I have more doubts. The method is not new, it has been described previously. The results presented here are correct to show if we have positive or negative results, but nothing more. I do not see any statistical data, curves showing how the signal changes for different concentration of CO2 (only 4 measurement points), is there any influence on the results coming from different gases, I am not sure if it is enough to concentrate only on a single wavelength. Moreover, the English writing is understandable, but could be improved. For sure the figures can be in Chinese.
Author Response

(The authors gave the same response as above.)

Round 2
Reviewer 1 Report
Dear Author,
thanks for the detailed response on my comments.
The updated manuscript is still full of corrections and difficult to check.
Please make sure that all the answers are properly reflected in the text.
Kind regards.
Author Response

(The authors gave the same response as above.)
